# Protective Effect of Carotenoid Extract from Orange-Fleshed Sweet Potato on Gastric Ulcer in Mice by Inhibition of NO, IL-6 and PGE_2_ Production

**DOI:** 10.3390/ph14121320

**Published:** 2021-12-17

**Authors:** Ji-Yeong Bae, Woo-Sung Park, Hye-Jin Kim, Ho-Soo Kim, Kwon-Kyoo Kang, Sang-Soo Kwak, Mi-Jeong Ahn

**Affiliations:** 1College of Pharmacy, Jeju Research Institute of Pharmaceutical Sciences and Interdisciplinary Graduate Program in Advanced Convergence Technology & Science, Jeju National University, Jeju 63243, Korea; jybae@jejunu.ac.kr; 2College of Pharmacy and Research Institute of Pharmaceutical Sciences, Gyeongsang National University, Jinju 52828, Korea; pws8822@gmail.com (W.-S.P.); black200203@gmail.com (H.-J.K.); 3Plant Systems Engineering Research Center, Korea Research Institute of Bioscience and Biotechnology, Daejeon 34141, Korea; hskim@kribb.re.kr (H.-S.K.); sskwak@kribb.re.kr (S.-S.K.); 4Division of Horticultural Biotechnology, Hankyong National University, Anseong 17579, Korea; kykang@hknu.ac.kr

**Keywords:** carotenoid, orange-fleshed sweet potato, gastric ulcer, anti-inflammatory activity

## Abstract

*Ipomoea batatas* (L.) Lam., Convolvulaceae is widely distributed in Asian areas from tropical to warm-temperature regions. Their tubers are known for their antioxidant, anti-bacterial, anti-diabetic, wound healing, anti-inflammatory, and anti-ulcer activities. The preventive and therapeutic effects of orange-fleshed sweet potato on gastric ulcers have not been investigated. In this study, the carotenoid extract (CE) of orange-fleshed sweet potato was found to protect against gastric ulcers induced by HCl/ethanol in mice. The anti-inflammatory and antioxidant activities of the carotenoid pigment extract were also evaluated as possible evidence of their protective effects. Administration of CE reduced gastric ulcers. Oral administration of CE (100 mg/kg) protected against gastric ulcers by 78.1%, similar to the positive control, sucralfate (77.5%). CE showed potent reducing power and decreased nitric oxide production in a mouse macrophage cell line, RAW 264.7, in a concentration-dependent manner. The production of the inflammatory cytokine interleukin-6 and prostaglandin E_2_ was also reduced by CE in a dose-dependent manner. The high carotenoid content of orange-fleshed sweet potato could play a role in its protective effect against gastric ulcers. This result suggests the possibility of developing functional products using this nutrient-fortified material.

## 1. Introduction

Gastric ulcer, also known as stomach ulcer, is the damage to the mucosal lining of the stomach with burning pain in the center of the abdomen [1]. The major factors in the development of gastric ulcers are the long-term use of nonsteroidal anti-inflammatory drugs, *Helicobacter pylori* infection, acetylsalicylic acid, alcohol consumption, smoking, and genetic factors [2]. According to the global burden of disease estimates, approximately 369,580 people were diagnosed with peptic ulcers (both gastric ulcers and duodenal ulcers) globally in 2020 [3]. Gastric ulcers are positively associated with gastric cancer [4]. It was reported that gastric cancer is the third most common cause of cancer associated death and the fifth highest frequent cancer in the world [5]. Moreover, South Korea has been reported to have the highest incidence of gastric cancer worldwide [6]. Various kinds of pharmaceutical medications are available for the relief of several gastric disorders, such as proton pump inhibitors, histamine type 2 receptor antagonists, gastric mucosal protectors, and a combination of antibiotics. However, the repeated use of these medications may result in unwanted side effects and may degrade the therapeutic benefits. Thus, there is a continuous need to develop alternative solutions for gastric ulcers from natural resources that are both effective and safe. In particular, the use of anti-ulcer food supplements has received particular attention and has been reported to have protective effects against gastric ulcers [7,8,9,10]. 

Sweet potato, *Ipomoea batatas* Lam (Convolvulaceae), has a long history of use as a staple food with high nutritional composition of starch, protein, vitamins, minerals, dietary fiber, and other bioactive compounds [11]. Roots and tubers of sweet potato play an important role in many underdeveloped countries, ranking fifth in essential crops cultivated in over 115 nations [12]. The anti-ulcer effects of white sweet potato powder have been reported regarding the mechanism of wound healing and radical scavenging activities [9,13]. Its antioxidant and radical scavenging activities might be able to reduce the potential hazards against free radicals [14]. Sweet potato is also recommended for stomach cancer patients because of its gastro-protective effects [15].

Orange-fleshed sweet potato has been developed for vitamin A deficiency (VAD), which provides high quantities of *β*-carotene, as well as attractive colors and sweet taste. Along with the World Health Organization report, over 190 million young children suffer from VAD due to poor intake of vitamin A-containing food. A medium level of orange-fleshed sweet potato variety (100 g) can meet the daily needs of vitamin A for an early age generation [16]. Orange-fleshed sweet potato is known for its anti-oxidative, anti-diabetic [17,18,19], and anti-obesity activities [20]. Regardless of the therapeutic effects of orange-fleshed sweet potatoes, the gastro-protective effects of carotenoid extract (CE) from orange-fleshed sweet potato have not been investigated. Thus, the gastric ulcer protective effect of CE from orange-fleshed sweet potato was examined using the HCl/ethanol-induced gastric ulcer mouse model. The antioxidant activity of CE was evaluated using reducing power, and its potential anti-inflammatory activity was studied using intracellular cytokine measurement as well as the LPS-induced nitric oxide reduction assay. The relationship between the protective effect of gastric ulcer and the measured bioactivities in our study enabled the development of functional products using orange-fleshed sweet potato.

## 2. Results

### 2.1. Chemical Profiles of CE from Orange-Fleshed Sweet Potato

Based on carotenoid analysis of CE from orange-fleshed sweet potato by HPLC, all-trans-ꞵ-carotene (92.2%), ꞵ-cryptoxanthin (2.6%), α-carotene (1.5%), zeaxanthin (1.2%), 13Z-ꞵ-carotene (0.9%), 9Z-ꞵ-carotene (0.8%), and lutein (0.7%) were identified from CE (Figure 1). All-trans-ꞵ-carotene was the dominant carotenoid [21]. The extraction yield of carotenoid extract from orange-fleshed sweet potato was 2.5% ± 0.1% (DW) and the total amount of carotenoids in the extract was 12.4 ± 0.4 mg/g DW.

### 2.2. Protective Effects of Carotenoid Extract on HCl/Ethanol-Induced Gastric Ulcer Model in Mice

Gastric ulcers were induced using hydrochloric acid and absolute ethanol, which is a popular tool for acute gastric mucosal devastation in a mouse model. Different from normal stomach (Figure 2A), the ulcer group showed the typical pathological patterns of mucosal damage, including dark reddish bands (Figure 2B), while the positive control (sucralfate) group and the carotenoid extract-treated group had a smaller lesion area with minor injury (Figure 2C,D). The ulcer area was assessed in the area of the inner gastric damage. The protective ratio was calculated based on the area of the negative control group’s value, which was counted as 100% damage. The dosage for oral administration (100 mg/kg) in mice was chosen according to a previous report [22]. With the administration of 0.15 M HCl in ethanol, the major gastric mucosal damage was induced with the hemorrhagic stripe form lesions (70.7 ± 3.2 mm^2^), which were significantly reduced to 15.5 ± 6.5 mm^2^ in the animals pretreated with CE at the dose of 100 mg/kg with the protective effect of 78.1% (Figure 2E). Sucralfate (100 mg/kg), used as a reference drug, also significantly reduced the gastric lesions by 77.5% (15.8 ± 6.0 mm^2^) when compared to the vehicle group (Figure 2E). The anti-ulcer activity of CE was similar to that of the positive control, sucralfate (Figure 2C–E). 

### 2.3. Antioxidant Activity Test with Reducing Power Assay

The reducing power was investigated by measuring the absorbance at 700 nm after mixing the CE with ferric compounds. A higher absorbance indicates a higher reducing power, which is related to antioxidant activity. In this assay, the yellow color of the original solution gradually changed to green or blue based on the reducing power of the samples. The application of antioxidants to the sample solution initiates the reduction from the ferric/ferricyanide complex to the ferrous ion, which can be evaluated by measuring the absorbance at 700 nm of Perl’s Prussian blue [23]. As shown in Figure 3, the reducing power of the CE increased in a concentration-dependent manner. The reducing capacity of CE was 84% compared to that of the reference compound, quercetin, at a concentration of 100 μg/mL.

### 2.4. Effect of CE on Intracellular NO Production in LPS-Stimulated RAW 264.7 Cells

Nitric oxide is involved in the modulation of gastric mucosal integrity [24], and the high concentration of NO produced by various cells plays a key role in the pathogenesis of inflammation [25]. Figure 4A shows the inhibitory activity of CE against LPS-stimulated NO production in RAW 264.7 cells to induce an inflammatory response by NO concentration. Based on the MTT assay, the experimental concentration range was maintained up to 100 μg/mL, as the cell viability was 91.9% at 100 μg/mL and 11.6% at 250 μg/mL (data not shown). NO production decreased in a dose-dependent manner at concentrations of 50 μg/mL and 100 μg/mL. At 100 μg/mL CE, it exhibited 55% inhibitory activity on NO production compared to the LPS-only treated group.

### 2.5. Effects of CE on LPS-Induced Cytokine Production in RAW 264.7 Cells

During inflammation, pro-inflammatory cytokines such as TNF-α, IL-6, IL-1ꞵ, and PGE_2_ promote the signaling pathways mediating inflammation-related disorders. After treatment with CE in LPS-stimulated RAW 264.7 cell, CE significantly inhibited the production of IL-6 and PGE_2_ in a concentration-dependent manner at a concentration range of 12.5–100 μg/mL (Figure 4B,C). At a concentration of 100 μg/mL, the production of IL-6 and PGE_2_ was reduced by 99% and 79.1%, respectively, but the level of TNF-α and IL-1ꞵ was not significantly reduced compared to the control group treated with LPS (data not shown).

## 3. Discussion

Gastric ulcers are a major digestive disorder, regardless of age, race, and regions in modern society. Although a variety of pharmaceutical drugs are commercially available for the treatment of gastric ulcers, serious adverse effects are still evident worldwide. In this respect, exploring gastric ulcer protective products from substances of natural origin has been of constant interest in research. The antiulcer properties of white sweet potato were reported by application of tuber flour water suspension to an ethanol-induced ulceration rat model [13]. Tuber flour suspension at 75 mg/kg showed a similar ulcer preventive effect as cimetidine at 100 mg/kg. They assumed that the carotenoids and polyphenol ingredients in white sweet potato could be associated with wound healing and ulcer preventive properties. In our previous research, we compared the carotenoid contents of white sweet potato and orange-fleshed sweet potato [21]. It was found that the orange-fleshed sweet potato showed significantly higher carotenoid content than the white sweet potato, which may be related to the gastric ulcer protective activity. Therefore, in this study, we evaluated the protective effects of the carotenoid extract (CE) from orange-fleshed sweet potato with an HCl/ethanol-induced gastric ulcer mouse model using a mechanism study including NO assay, intracellular cytokine measurement, and reducing power.

The HCl/ethanol-induced gastric ulcer model is associated with biological activities such as NO inhibition, pro-inflammatory cytokines, and intracellular oxidative stress [26]. Ethanol is a widely recognized agent of gastric ulceration that promotes mucosal epithelial cell apoptosis, oxidative stress in gastric tissue, and inflammatory reactions [27]. In this study, CE from orange-fleshed sweet potato exhibited significant anti-gastric ulcer activity in the HCl/EtOH-induced gastric ulcer mouse model. Gastric ulcers are known to be closely related to inflammatory processes in the gastric mucosa [8]. NO is known as a mediator and regulator of inflammatory responses and is involved in the modulation of gastric mucosal integrity [24,28]. The results of the present study demonstrated that CE significantly inhibited LPS-induced NO production in a concentration-dependent manner (Figure 4A).

The effects of CE on LPS-induced cytokine production in RAW 264.7 cells were evaluated for their anti-inflammatory activities. RAW 264.7 cells were chosen for their anti-inflammatory effects as they release pro-inflammatory cytokines by LPS stimulation [29]. LPS-induced RAW 264.7 cells produce a number of inflammatory mediators such as NO, tumor necrosis factor-α (TNF-α), interleukin-6 (IL-6), interleukin-1β, (IL-1β), and PGE_2_ through nuclear factor-kappa B (NF-κB) and the mitogen-activated protein kinase (MAPK) pathways. It is known that the levels of these mediators can be a key evidence of the inflammation process [30,31]. Activated NF-κB can move into the nucleus to promote the expression of pro-inflammatory genes and be involved in several inflammatory diseases [32]. MAPKs consist of extracellular-signal-regulated kinase ERK1/2, p38 MAP kinase, and c-Jun N-terminal kinases (JNK) [33]. Once the MAPKs are activated, the p38, JNK, and ERK transcription factors are phosphorylated and activated, initiating the inflammatory response [34]. Among the pro-inflammatory cytokines, IL-6 plays a key role in acute and chronic inflammatory responses by stimulating immune cells, macrophages, and acute-phase protein synthesis. IL-6 upregulation has been observed in several inflammatory disorders [35]. PGE_2_ also induces acute and chronic inflammation through master cell activation and helper T1 (Th1) cell differentiation of helper T1 (Th1) cells, Th17 cells, and IL-22 production [36]. In this study, CE reduced both IL-6 and PGE2 in a dose-dependent manner (Figure 4B,C), while the significant reduction of IL-1β and TNF-α level was not observed in ELISA assay. Two major carotenoids of CE, *β*-carotene and lutein are known for anti-inflammatory carotenoids via inhibition of the NF-κB and MAPK signaling pathways [31,37]. Further study on the mechanism of anti-inflammatory effect on CE can be revealed through RT-PCR and Western blot assays for the mRNA and protein expression of iNOS, COX-2 as well as IL-6, IL-1β, and TNF-α.

Reactive oxygen species (ROS) are responsible for human disorders such as inflammation, neurodegenerative disorders, gastrointestinal inflammation, and ulcers [38]. ROS are generated by the metabolism of arachidonic acid, macrophages, and smooth muscle cells, contributing to gastric mucosal damage. In this study, CE showed potent antioxidant activity, representing the contribution of protective roles to the oxidative stress induced by ethanol in the stomach mucosa. In particular, the high reducing power of CE in the FRAP assay (Figure 3) suggests that the antioxidant activity may be one of the mechanisms of its gastro-protective and anti-inflammatory properties, because both ulcerous and inflammatory processes are related to oxidative stress [39]. 

Carotenoids are lipophilic pigments found in plants and microorganisms and are widely distributed in nature with numerous functions such as photo-protection and sexual attraction due to their vivid color [40,41]. Previous research on the gastro-protective effects of carotenoids showed various results depending on properties such as antioxidants. Astaxanthin from *Paracoccus carotinifaciens* has been reported to have protective effects in a murine gastric ulcer model induced by hydrochloride/ethanol or acidified aspirin [22] along with antioxidant properties. Lutein is known for its protective effects against gastric ulcers [42] due to its ability to inhibit oxidative stress produced by alcohol. Carotenoids and anacardic acid-enriched water extracts from cashew apple byproducts also have gastro-protective effects on acetylsalicylic acid-induced gastric lesions in rats [43]. *ꞵ*-carotene is known for its antioxidant and anti-ulcer effects against indomethacin-induced gastric ulceration [44]. In this study, *ꞵ*-carotene, which is the main carotenoid in orange-fleshed sweet potato, plays a key role in its gastro-protective effects due to its antioxidant mechanisms. 

## 4. Materials and Methods

### 4.1. Chemicals

Sucralfate, lipopolysaccharide (LPS), ferric chloride, trichloracetic acid (TCA), sodium nitrite, potassium ferricyanide, dimethyl sulfoxide (DMSO), *N*-(1-naphthyl) ethylenediamine dihydrochloride, phosphoric acid, and sulfanilamide were purchased from Sigma-Aldrich, Inc. (St. Louis, MO, USA). Dulbecco’s modified Eagle’s medium (DMEM), the antibiotic mixture (penicillin-streptomycin), and fetal bovine serum (FBS) were purchased from Hyclone (South Logan, UT, USA). All chemicals not listed were grade of reagent.

### 4.2. Plant Material

Orange-fleshed sweet potato (cv. Sinhwangmi) was obtained from the Bioenergy Crop Research Center, National Institute of Crop Science, Korea. The sweet potatoes used in this study were cultivated in the same batch and harvested after four months of planting. The tubers were washed in tap water, dried, and lyophilized after the removal of peels. The voucher sample (PGSC No. 1301) was preserved in the Herbarium of the College of Pharmacy, Gyeongsang National University.

### 4.3. Sample Preparation

The carotenoid extract (CE) was prepared from a freeze-dried materials of orange-fleshed sweet potato (cv. Sinhwangmi), extracted in sonicator for three times, 10 min each in acetone. The extract was centrifuged at 5700× *g* at 4 °C for 10 min (5430R, Eppendorf, Hamburg, Germany), and the supernatant was condensed using a rotary evaporator. The condensed extract was dried in a N_2_ dryer and prepared for the assay. All extraction procedures were performed under subdued light to avoid pigment degradation.

### 4.4. Liquid Chromatography Diode Array Detector Instrumental Condition

For the analysis of carotenoid composition in orange-fleshed sweet potato, HPLC-DAD method was applied according to our previous report [20]. Briefly, 1 mg of each standard was dissolved in CH_2_Cl_2_ (0.01% BHT, 10 mL) to make stock solution. Working calibration solutions (0.025~50 μg/mL) were then prepared by diluting stock solution of the external standards. All carotenoid standards were acquired from CaroteNature (Lupsingen, Switzerland). HPLC analysis was done by using an Agilent 1100 HPLC system (Santa Clara, CA, USA). The carotenoid extract and reference compounds were taken in the volume of ten microliters and injected onto a YMC C30 carotenoid column (3 µm, 4.6 × 250 mm, Japan) with the mobile phase of solvent A (methanol/*tert*-butyl methyl ether/water (81:15:4, *v*/*v*)) and solvent B (methanol/*tert*-butyl methyl ether/water (6:90:4, *v*/*v*)) using a step gradient mode of 100% solvent A for the first 15 min, then 100% solvent A to 100% solvent B over the next 35 min. The following 10 min was set up to equilibrate the column to its initial state. The column temperature was 22 °C and flow rate was 0.7 mL/min. The eluent was detected at 450 nm using an UV-Visible detector. The Chemstation software (Santa Clara, CA, USA) was used to operate LC system. Under these chromatographic conditions, standard carotenoids were separated with the *t*_R_ (min) values of 23.3 for lutein, 25.7 for zeaxanthin, 32.3 for *ꞵ*-cryptoxanthin, 35.1 for 13*Z*-*β*-carotene, 35.9 for α-carotene, 38.2 for all-*trans*-*β*-carotene, and 39.7 for 9Z-*β*-carotene (Figure 1).

### 4.5. Animals

Four-week-old male ICR mice in the range of 24−28 g weight (Koatech, Korea) were used in the gastric ulcer protective experiment. Each mouse was kept in an air-conditioned chamber at 24 ± 2 °C under a 12 h light/dark cycle during one week for the acclimatization. Each mouse was used for one time per each experiment. All procedures regarding animal care and treatment conformed to the Animal Care Guidelines of the Animal Experiment Committee of Gyeongsang National University (IACUC No. GNU-120508-M0018).

### 4.6. HCl/Ethanol-Induced Gastric Ulcer

The gastric ulcer protective activity against hydrochloric acid/ethanol was estimated using a previously reported method with slight modifications [8,45]. After a fasting period of 16 h, the mice were randomly divided into four groups, consisting of five mice per group. The sham-operated and vehicle-treated groups were given 0.2 mL of 0.25% sodium carboxymethylcellulose (CMC-Na) (Samchun Chemicals, Korea) solution. Positive control group was pre-treated with 0.2 mL of sucralfate (100 mg/kg, *p.o.*) diluted in 0.25% CMC-Na solution. The other group was treated with CE in 0.25% CMC-Na solution (100 mg/kg, *p.o.*). All the materials for oral administration to animals were kept in a dark freezer and prepared one hour before the treatment. The mice except ones belonging to the sham-operated group were administered 150 mM HCl/ethanol (0.2 mL/25 g body weight) after 1 h of sample treatment. One hour after treatment with HCl/ethanol, the mice were euthanized by carbon dioxide inhalation. The stomachs were then transferred and fixed with 1% formalin for 30 min. The stomachs were then operated through the greater curvature and washed with saline solution. Ulcer evaluation was performed by measuring the area of the gastric lesion using image analysis software (Isolution Lite, IMT i-solution Inc., Vancouver, BC, Canada) and expressed as mean ± standard error of the mean (S.E.M).

### 4.7. Reducing Power Assay

The reducing power of the functional pigment extract was measured using the previously reported method with slight modification [22]. Specifically, 2.5 mL of 2 M phosphate buffer (pH 7.0) and 2.5 mL of 1% (*v*/*v*) potassium ferricyanide (Sigma, St. Louis, MO, USA) were added to 1 mL of the functional pigment extract at 50 °C and reacted for 20 m. Next, 2.5 mL of 10% (*w*/*v*) trichloracetic acid was added, and centrifuged for 10 m, and 2.5 mL of the supernatant was taken and same portion of distilled water, 0.5 mL of 0.1% (*w*/*v*) ferric chloride were mixed. After incubating this solution for 10 m, absorbance was measured at 700 nm.

### 4.8. Cell Culture

RAW 264.7 cells were purchased from The Korean Cell Line Bank (KCLB, Korea). The murine macrophage cell line RAW 264.7 was cultured at 37 °C and 5% CO_2_ in DMEM supplemented with 1% penicillin/streptomycin and 10% fetal bovine serum. The cells were sub-cultured every 2–3 days, and cells within 20 passages were used in the experiment.

### 4.9. Cell Viability Assay

RAW 264.7 cells were distributed into 96-well plates at a density of 1 × 10^5^ cells/well and incubated for 24 h. The culture medium was replaced with fresh culture medium containing each sample solution at final concentrations of 5, 10, 50, 100, 250, and 500 μg/mL, and cultured for four days. After cultivation, the medium was removed from each plate, and MTT solution containing 3-(4,5-dimethylthiazol-2-yl)-2,5-diphenyl-2*H*-tetrazolium bromide at a concentration of 0.5 mg/mL in PBS was added to 100 μL per well and incubated at 37 °C for 4 h. The plates were centrifuged at 6800× *g* for 5 min, followed by removal of the medium and dissolution of formazan crystals on the bottom using DMSO. The absorbance was measured using a Victor X5 microplate reader (PerkinElmer, Waltham, MA, USA) at 570 nm. Cell viability was calculated by subtracting the mean values without assay reagent (sham) from those with MTT solution and was expressed as a percentage of the control (cell viability, %).

### 4.10. Intracellular Nitric Oxide Production

RAW 264.7 cells were plated at a density of 1 × 10^5^ cells/well in 96-well plates and incubated for 24 h. These cells were treated with each sample solution at final concentrations of 5, 10, 50, and 100 μg/mL for 3 h before exposure to LPS (1 ng/mL) for stimulation. The nitrite levels in the culture media were measured by adding Griess reagent to promote nitric oxide (NO) production, followed by a 24 h incubation [46]. The calibration curve was constructed with 100 μM sodium nitrite as the highest concentration. Absorbance was measured at 544 nm using a Victor X5 multi-label plate reader (PerkinElmer, USA). The blank was fresh culture medium in all experiments, and the value was used in the calculation correction. Nitrite levels in the samples were determined using a standard sodium nitrite calibration curve.

### 4.11. PGE_2_ Production

PGE_2_ in the cell culture was measured using a prostaglandin E_2_ parameter assay kit purchased from R&D Systems (Minneapolis, MN, USA). RAW 264.7 cells were treated with CE with 100 ng/mL LPS, followed by 18 h of incubation. In a 96-well plate coated with goat-anti-mouse, 150 μL of the cell culture solution was added, and 50 μL of primary antibody solution was added. 50 μL of PGE_2_ conjugation was added and reacted for 2 h, washed four times with washing buffer, and 200 μL of the substrate solution was treated for 30 min. The absorbance was measured at 540 nm after treatment with 1000 μL of the stop solution.

### 4.12. TNF-α, IL-1β and IL-6 Production

Cytokine production was measured by enzyme-linked immunosorbent assay (ELISA) purchased from R&D Systems (Minneapolis, MN, USA). Cells were treated with CE and incubated for 1 h. Then, LPS was added at a concentration of 100 μg/mL and incubated for 18 h. The cell culture solution was diluted to an appropriate concentration, and 50 μL of each was added to a 96-well plate coated with cytokine, and 50 μL of primary antibody solution was added to react for 2 h at room temperature. After washing three times with washing buffer, 100 μL of cytokine conjugation to be measured was treated in each well and reacted at room temperature for 2 h. Washing again three more times with washing buffer, 100 μL of substrate solution was treated for 30 min in dark condition. After washing again three times with washing buffer, 100 μL of stop solution was added, and absorbance was measured at 540 nm.

### 4.13. Statistical Analysis

The statistical significance of the differences among samples was statistically evaluated using one-way analysis of variance (ANOVA). The values were considered statistically significant at *p*-values less than or equal to 0.05.

## 5. Conclusions

Pretreatment with CE from orange-fleshed sweet potato relieved the gastric mucosal injury caused by hydrochloric acid/ethanol through anti-inflammatory, antioxidant, and pro-inflammatory cytokine modulation. CE showed potent reducing power and reduced nitric oxide production in a mouse macrophage cell line, RAW 264.7, in a concentration-dependent manner. The production of the inflammatory cytokines interleukin-6 and prostaglandin E_2_ was also reduced by CE in a dose-dependent manner. These findings contribute to the pharmacological validation of orange-colored sweet potatoes with anti-inflammatory properties and the possibility of gastro-protective effects not only using medication but also food sources. Further phytochemical and pharmacological investigations are still needed to better elucidate the exact mechanism of action of the pigment extract -from color-fleshed sweet potato.

## Figures and Tables

**Figure 1 pharmaceuticals-14-01320-f001:**
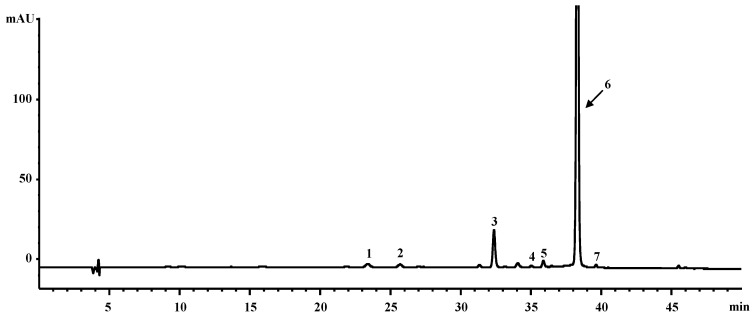
HPLC chromatogram of carotenoid extract (CE) from orange-fleshed sweet potato. **1**, lutein; **2**, zeaxanthin; **3**, ꞵ-cryptoxanthin; **4**, 13Z-ꞵ-carotene; **5**, α-carotene; **6**, all-trans-β-carotene; **7**, 9Z-ꞵ-carotene.

**Figure 2 pharmaceuticals-14-01320-f002:**
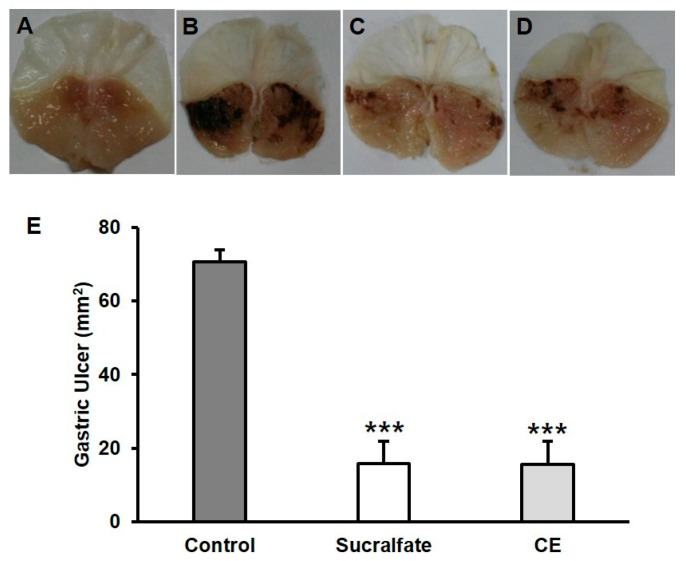
The protective effects of carotenoid extract (CE) from orange-fleshed sweet potato on HCl/ethanol-induced gastric ulcer model in mice. (**A**), normal stomach; (**B**), glandular stomach treated with vehicle; (**C**), the stomach of positive control group treated with sucralfate, 100 mg/kg; (**D**), the stomach of test group treated with CE, 100 mg/kg, *p.o.,* 1 h before administration of 150 mM HCl/ethanol; (**E**) Determination of the gastric ulcer area is expressed in the unit of mm^2^. Data are presented as mean ± standard error of the mean (S.E.M). *** *p* < 0.001, significantly different from gastric ulcer group (**B**).

**Figure 3 pharmaceuticals-14-01320-f003:**
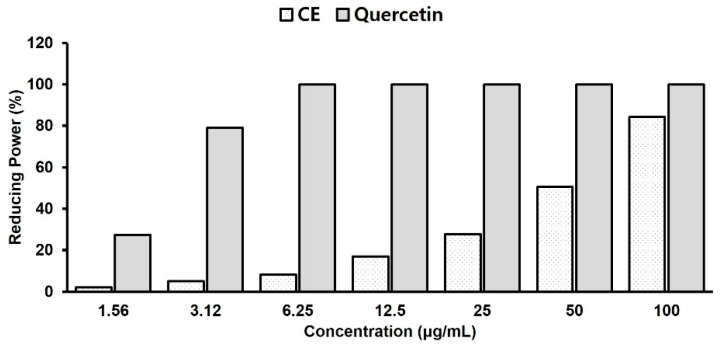
Total antioxidant activity measured by ferric reducing antioxidant power (FRAP) assay for the carotenoid extract (CE). Quercetin is used as a reference compound (concentration range; 1.6–100 µg/mL). The data are presented as mean ± standard deviation.

**Figure 4 pharmaceuticals-14-01320-f004:**
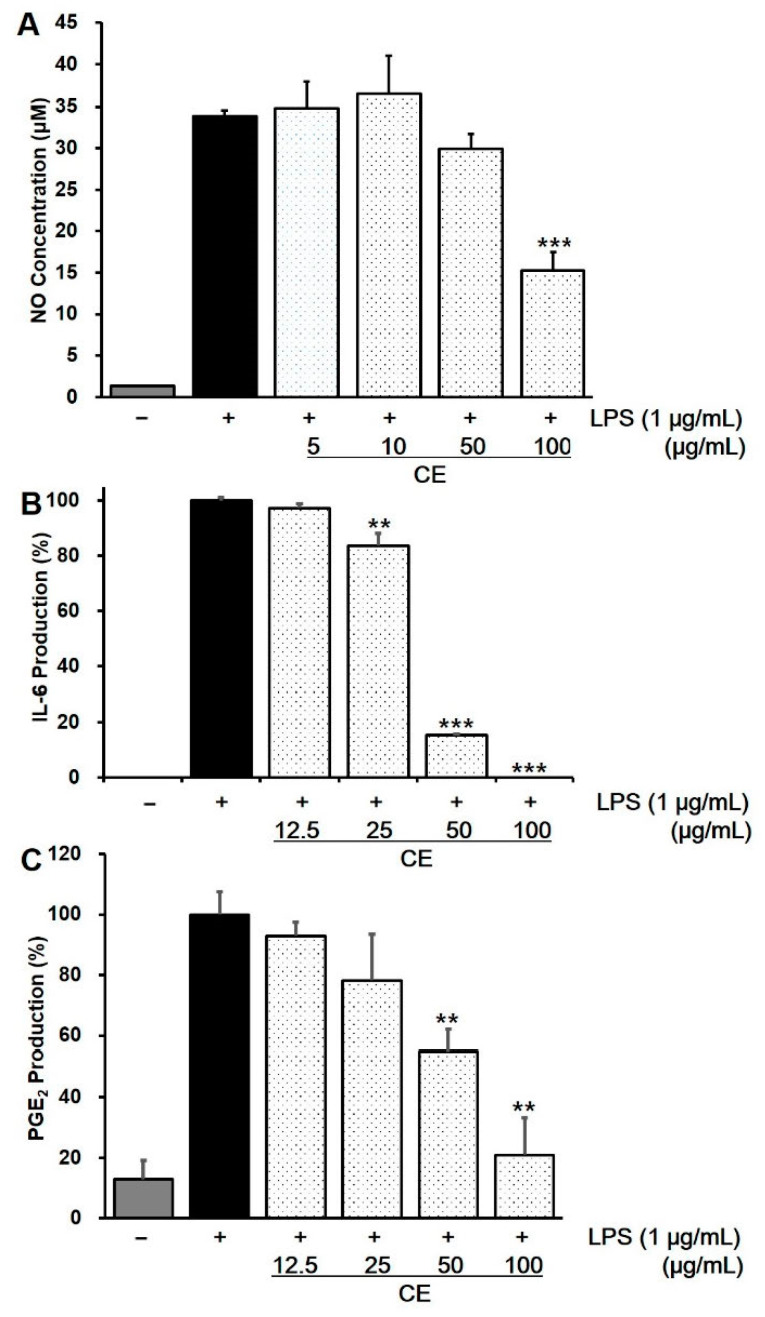
Effect of carotenoid extract (CE) on NO production (**A**); IL-6 production (**B**); PGE_2_ production (**C**) in LPS-stimulated RAW 264.7 cells. ** *p* < 0.01, *** *p* < 0.001 vs. LPS-treated control.

## Data Availability

Data is contained within the article.

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
