# Peer review of "Protective Effect of Carotenoid Extract from Orange-Fleshed Sweet Potato on Gastric Ulcer in Mice by Inhibition of NO, IL-6 and PGE2 Production"

_pharmaceuticals, 2021, doi:10.3390/ph14121320_

Round 1

Reviewer 1 Report

The reviewed manuscript entitled "Protective effect of carotenoid extract from orange-fleshed sweet 2 potato on gastric ulcer in mice by inhibition of NO, IL-6 and 3 PGE2 production” is an interesting and original paper, where the authors determine the anti-inflammatory, antioxidant activities of the carotenoid pigment extract and its protective effect against gastric ulcers.

The introduction provides a good and generalized background of the topic. Material and Methods are appropriate and have been described. Results and discussion are clearly explained and the conclusions are mostly well supported by the results. 

Minor  points I found that  deserve attention by the Authors: 

the Reviewer suggests a revision of the issue regarding the  cell signalling pathway able to suppress the proinflammatory molecules by inhibition of phosphorylated extracellular signal-regulated kinase (ERK), phosphorylated c-Jun nterminal kinase (JNK) expression, nuclear factor kappa B (NF-kB) activation, and finally  epigenetic modification involved in anti-inflammatory effects. This should at least be included in the introduction and / or discussion.

 doi.org/10.3390/md19100531   10.1016/j.fct.2014.08.005   10.1186/s13148-020-00930- doi.org/10.1111/bph.14888

Author Response

Response to reviewer #1: 

Thank you very much for your careful and thorough reading of this manuscript and for the thoughtful comments and constructive suggestions, which helps to improve the quality of this manuscript. We have considered your comments (Original Reviewer’s Comments, ORC) very closely in revising this manuscript as follows.

ORC: The reviewer suggests a revision of the issue regarding the cell signaling pathway able to suppress the pro-inflammatory molecules by inhibition of phosphorylated extracellular signal-regulated kinase (ERK), phosphorylated c-Jun n terminal kinase (JNK) expression, nuclear factor kappa B (NF-kB) activation, and finally epigenetic modification involved in anti-inflammatory effects. This should at least be included in the introduction and / or discussion.

  • In the Discussion Section, the inflammatory cell signaling pathways were included to explain general response against LPS-stimulus in macrophage cells (lines 194-204).

Thank you very much. One of your suggested articles was added as a reference (#41) for the inflammatory or anti-ulcer activity of carotenoids.

Reviewer 2 Report

The subject is interesting and of real practical application.

There are minor spelling errors that can be checked and corrected.

The results are very concisely and well presented, but the discussion for me it seems like a list of facts without linking between the results of authors and the existing data. There are not highlighted the authors' results in the context of literature data. I suggest tobe  reconsidered this part and to be discussed the results by comparison with the literature data.

At materials and methods the HPLC method conditions is not described. Please complete with these data.

At the same chapter there are determined also other pro-inflammatory cytokines (TNF-alpha, IL-1beta), but at results are presented only IL-6. Please clarify this situation.

Due by the observation regarding the discussions I will indicate major revision of paper before publishing.

Author Response

Response to reviewer #2: 

Thank you very much for your careful and thorough reading of this manuscript and for the thoughtful comments and constructive suggestions, which helps to improve the quality of this manuscript. We have considered your comments (Original Reviewer’s Comments, ORC) very closely in revising this manuscript as follows.

ORC: The subject is interesting and of real practical application. There are minor spelling errors that can be checked and corrected. The results are very concisely and well presented, but the discussion for me it seems like a list of facts without linking between the results of authors and the existing data. There are not highlighted the authors' results in the context of literature data. I suggest to be reconsidered this part and to be discussed the results by comparison with the literature data.

  • Minor spelling errors were checked and corrected. In the Discussion Section, the results were discussed by comparison with other literature data (lines 211-213 and 401-410).

ORC: At materials and methods the HPLC method conditions is not described. Please complete with these data.

  • The concrete HPLC method conditions were added in the Materials and Methods Section (lines 475-484).

ORC: At the same chapter there are determined also other pro-inflammatory cytokines (TNF-alpha, IL-1beta), but at results are presented only IL-6. Please clarify this situation.

  • In the Results Section, the results on TNF-α and IL-1β were mentioned as “the level of TNF-α and IL-1êžµ was not significantly reduced compared to the control group treated with LPS (data not shown).”. The related discussion was also added in the Discussion Section (lines 210-215).

Thank you.

Round 2

Reviewer 2 Report

Thanks to take in consideration my observations. I consider that the paper in revised form can be published.